# A Comparison of E-Cigarette Use Patterns and Smoking Cessation Behavior among Vapers by Primary Place of Purchase

**DOI:** 10.3390/ijerph16050724

**Published:** 2019-02-28

**Authors:** Greta Hsu, Anthony C. Gamst, Yue-Lin Zhuang, Tanya Wolfson, Shu-Hong Zhu

**Affiliations:** 1Graduate School of Management, University of California, Davis, Davis, CA 95616, USA; grhsu@ucdavis.edu; 2Department of Mathematics, University of California San Diego, La Jolla, CA 92093-0112, USA; acgamst@ucsd.edu; 3San Diego Supercomputer Center, University of California San Diego, La Jolla, CA 92093-0505, USA; twolfson@ucsd.edu; 4Moores Cancer Center, University of California San Diego, La Jolla, CA 92093-0905, USA; y1zhuang@ucsd.edu; 5Department of Family Medicine and Public Health, University of California San Diego, La Jolla, CA 92093-0905, USA

**Keywords:** electronic cigarettes, purchase channels, Marketing and regulation of nicotine-containing products, smoking cessation, regulation, vape shops, retail, internet

## Abstract

Background: E-cigarettes are purchased through multiple channels, including general retail, online, and specialty smoke and vape shops. We examine how e-cigarette users’ primary purchase place relates to e-cigarette use and smoking cessation behaviors. Methods: Probability-based samples of the U.S. population who were current e-cigarette users were surveyed in 2014 (N = 879) and 2016 (N = 743), with responses combined for most analyses. E-cigarette use and smoking cessation behaviors were compared across users’ primary purchase place. Results: Higher percentages of vape shop (59.1%) and internet (42.9%) customers were current daily users of e-cigarettes compared to retail (19.7%) and smoke shop (23.2%) customers (*p*-values < 0.001). Higher percentages of vape shop (40.2%) and internet (35.1%) customers were also former smokers, compared to 17.7% of retail and 19.3% of smoke shop customers (*p*’s < 0.001). Among those smoking 12 months prior to survey, smoking cessation rates were higher for vape shop (22.2%) and internet customers (22.5%) than for retail customers (10.7%, *p* = 0.010 and *p* = 0.022, respectively), even though retail customers were more likely to use FDA-approved smoking cessation aids. The percentage of customers purchasing from vape shops increased from 20.4% in 2014 to 37.6% in 2016, surpassing general retail (27.7%) as the most likely channel in 2016. Conclusions: E-cigarette customers differed in significant ways by channels of purchase, most notably in their smoking cessation behaviors. Previous population studies have relied mostly on retail channel data, which accounted for less than 30% of all products sold by 2016. Future studies of e-cigarette use should consider a broader set of channels.

## 1. Introduction

E-cigarettes can be purchased through several channels in the U.S., including general retail locations (e.g., supermarkets, convenience stores), online stores, and specialty smoke and vape shops. Each appear to be major outlets for e-cigarettes, although sales through internet and specialty shops have generally not been well-tracked and channels are often combined differently within different industry surveys [1,2]. For example, Coleman et al. estimated that, in 2013–2014, 29% of established e-cigarette users purchased e-cigarette products from convenience stores/gas stations, while 56% purchased from a tobacco/smoke or vape shop, and 14% from the internet [3]. Braak et al. found that in 2016, roughly 43.3% of U.S. e-cigarette users purchased vaping products through a vape shop, 26.8% purchased online, and 29.9% purchased through “other retail” channels (including tobacco specialty shops, supermarkets, pharmacy, and convenience stores) [4]. In recent years, a proliferation of vape shops in the U.S. further indicates their increasing position as a source of e-cigarette purchases [5,6].

Gaining an accurate picture of e-cigarette sales across channels is important for understanding this young industry’s evolution. It is also important, from a public health viewpoint, to understand how each channel relates to e-cigarette users’ behavior with regards to cigarette smoking and intentions to quit. Qualitative evidence suggests different purchase channels expose e-cigarette consumers to different types of information about electronic nicotine delivery systems (ENDS). Vape shops appear particularly distinctive in this regard. Many vape shop owners and staff are former cigarette smokers who—despite lack of formal training—provide testimonials, guidance, and informal counseling to customers interested in smoking cessation [5,7,8,9,10]. Vape shops also encourage the development of a community of e-cigarette enthusiasts by hosting social events and providing a venue for users to socialize and share information [7,10,11,12]. The different roles vape shops and other outlets play in tobacco control, whether positive or negative, deserves study.

This study has two objectives. First, it aims to develop understanding of whether e-cigarette users’ primary place of purchase is related to their e-cigarette use patterns, smoking quit attempts, and likelihood of smoking cessation. To date, there has been little research comparing user behaviors across multiple e-cigarette purchase channels. Importantly, the present study distinguishes between vape shops and smoke shops that carry vaping products. While industry surveys commonly combine the two, qualitative studies suggest that vape shops often present different beliefs about cigarettes (such as anti-smoking views) compared to smoke shops that carry both vaping and smoking products [5,9,13]. The two forms may thus be associated with different types of smoking behavior.

A secondary aim is to obtain a more complete picture of the distribution of e-cigarette sales across the different major channels and how this evolved from 2014 to 2016. Market research firms such as Nielsen focus on traditional retail channels such as convenience stores, g’as stations, and drug stores and do not track e-cigarette sales conducted online or through specialty tobacco and vape shops [2]. Given that these appear to be major channels for e-cigarette purchases in the U.S. [3,4], this study instead tries to obtain relative estimates for sales directly from a representative sample of e-cigarette users.

This study is based on two surveys of e-cigarette users—the first conducted in 2014 and the second in 2016. Both surveys are based on a nationally representative sample of U.S. adults.

## 2. Materials and Methods

### 2.1. Data Source

The 2014 and 2016 surveys were both drawn from GfK’s KnowledgePanel, an ongoing probability-based panel that relies on address-based sampling from a sample frame of residential addresses that covers approximately 97% of U.S. households [14]. This panel was first developed in 1999 by Knowledge Networks, an online consumer product and service research firm acquired by the market research organization GfK. This sample includes individuals with unlisted phone numbers and without landlines or Internet access; the latter were provided a computer with WiFi to complete the survey. We increased sampling efficiency by oversampling ever smokers (because the majority of e-cigarettes users are ever smokers [15,16]). Out of 9334 invited adults, 8619 completed the 2014 survey, yielding a 92.3% response rate. Out of 11,509 people, 10,314 completed the 2016 survey—an 89.5% response rate. Because our aim was to gather information on current e-cigarette users’ purchase and smoking-related behaviors, we only included in our sample those respondents who reported their primary place of purchase. Our analyses focus on 879 current e-cigarette users from the 2014 survey and 743 from the 2016 survey.

This research was approved by the University of California, San Diego’s Institutional Review Board (IRB #111664). Participants provided informed consent.

### 2.2. Measures

‘E-cigarette users’ were defined as respondents who answered using e-cigarettes “every day” or “some days” to the question: “Do you currently use e-cigarettes every day, some days or not at all?” To determine e-cigarette users’ ‘primary purchase channel’, they were asked, “Where do you usually buy your e-cigarettes or e-liquid?” Responses to the question are: “Internet,” “Pharmacy,” “Gas station/convenience store,” “Grocery store/supermarket,” “Smoke shop,” “E-cigarette/vape shop,” “Other,” and “I have never bought e-cigarettes.” Retail locations include “Pharmacy,” “Gas station/convenience store,” and “Grocery story/supermarket.” Those who answered “other” or “I have never bought e-cigarettes” were excluded from the analysis.

We defined an ‘ever smoker’ as having smoked ≥100 cigarettes in one’s lifetime. Ever smokers were further broken into two groups: ‘current smokers’ (ever smoker, smoked every day or some days at time of survey) and ‘former smokers’ (did not smoke at time of survey). All ever smokers were asked whether they were smoking every day or some days 12 months prior to the time of survey.

To determine e-cigarette users’ current e-cigarette model in 2014, we presented respondents with photos of different systems and asked a series of questions. Each question was accompanied by photos with descriptive captions to illustrate the types of e-cigarette systems referred to. ‘Closed system users’ were those who answered “Yes” to one of the following two questions:

1. “Do you currently use disposable e-cigarettes?”; caption: “Looks like a cigarette; one piece.”

2. “Do you currently use a 2-piece (or 3-piece) e-cigarette similar to the ones shown below (not (personal vaporizer) PV or (advanced personal vaporizer) APV)?”; caption: “2 or 3 piece; consists of battery and cartomizer (or cartridge and atomizer).”

*Open system users* were those who answered “Yes” to one of the following two questions:

1. “Do you currently use a PV or APV e-cigarette (e.g., eGo) similar to the ones shown below?”; caption: “Personal vaporizer. Customizable. May be used with tanks, cartomizers, clearomizers, or drip tips.”

2. “Do you currently use a MOD or handmade e-cigarette?”; caption: “Personal vaporizer. Endlessly customizable. May be used with tank devices, cartomizers, clearomizers, or drip tips.”

In 2016, the questions for e-cigarette model use were more straightforward. E-cigarette users were presented with two sets of photos and captions:

1. “Closed System—any disposable or pre-filled cartridge system. You do not add your own e-liquid.”

2. “Open system—any system you refill using bottles of e-liquid.”

Then, they were asked, “Currently, which type of e-cigarette do you use MOST OFTEN?” Their e-cigarette model used was determined by their response: “Closed system”, Open system”, or “Use both closed and open systems equally.”

E-cigarette users who reported being smokers 12 months before the survey were asked whether they attempted to quit smoking in the past 12 months. A quit attempt was defined as having quit smoking for ≥24 hours. Among those who reported a quit attempt in the past 12 months, we asked, “When you quit smoking the LAST TIME, did you use any of the following to help you quit? Nicotine patches, nicotine gum, nicotine lozenges, nicotine spray or inhaler, Zyban/Wellbutrin (Bupropion), Chantix (Varenicline), e-cigarettes (vapes, vape pen, hookah pen, e-hookah), Smokeless tobacco (chew, dip, snus).” Respondents indicated yes or no to each. The first four options were collapsed into NRT (nicotine replacement therapy) use. We defined users who quit smoking as respondents who were smoking 12 months ago but reported quitting by the time of survey.

### 2.3. Statistical Analysis

In most analyses, we combined the two surveys when summarizing and comparing responses to survey questions. While 58.5% of included respondents were exclusive to either the 2014 or 2016 survey, 41.5% participated in both the 2014 and 2016 surveys. Given this substantial overlap, the surveys could not be considered independent, with negligible error due to serial dependence. Our approach deals with the serial dependence of respondents who participate in both surveys by transforming the data into two independent surveys via random splitting of the dependent sample, averaging over the large number of such splits and accounting for the resultant additional variance. This approach produces unbiased estimates.

More specifically, we sampled from the binomial distribution to place each respondent at random into exactly one of the surveys to which they responded. This respondent’s weight in the other survey was set to 0. Survey weights were then recomputed so that the total weight of each survey equaled the total number of respondents for that year. Overall weighted proportions for each survey item of interest were computed, as well as the differences in proportions for the items to be compared and the standard errors associated with each proportion and each difference. We then repeated this procedure 2000 times to obtain 2000 sets of weighted proportions, differences, and associated standard errors. Those data were used to compute averaged weighted proportions and both conditional and unconditional variance components. Confidence intervals and *p*-values for differences in proportions were computed around each weighted statistic using those variance components.

## 3. Results

Table 1 shows demographics of survey respondents. Overall, the differences across these four purchase channels were modest. The biggest difference was between vape and smoke shop customers, in terms of education. Vape shops had a higher percentage of high-education customers compared to smoke shops (50.1% versus 39.2%, *p* < 0.05).

Vape shop e-cigarette users had the lowest percentage of users in the youngest age bracket (18–24), 3.2%, compared with 12.0% for retail channels (*p* < 0.01), with 12.9% for smoke shops (*p* < 0.01), and 15.5% for internet (*p* < 0.001) users. They had higher percentage in the 25–34 age bracket, 39.3%, compared with retail, 24.5% (*p* < 0.01) and smoke shop, 24.0% (*p* < 0.01)

Table 2 is divided into two halves. The top half includes all e-cigarette users while the bottom includes only those e-cigarette users who reported that they were smoking 12 months before the survey.

The top half of Table 2 shows that most current e-cigarette users were ever smokers. Vape shop customers had the highest percentage of ever smokers (94.7% = 54.5% current smokers + 40.2% former smokers). This percentage was higher than that of internet (83.8% = 48.7% + 35.1%; *p* < 0.001) but did not differ from retail customers (91.4% = 73.7% current smokers + 17.7% former smokers; *p* = 0.36) and smoke shop customers (89.7% = 70.4 current smokers + 19.3% former smokers; *p* = 0.12). Vape shop and internet customers were more likely to be former smokers, 40.2% and 35.1%, respectively—each a significantly higher percentage when compared to the 17.7% of retail and the 19.3% of smoke shop customers who were former smokers (*p*’s < 0.01).

E-cigarette usage pattern differed significantly across groups. A significantly higher percentage of vape shop (59.1%) and internet (42.9%) customers reported daily use of e-cigarettes relative to retail (19.7%) and smoke shop (23.2%) customers (*p*’s < 0.001).

The sharpest difference across these groups was in their choice of e-cigarette system. Vape shop customers overwhelmingly preferred open-system designs, with 92.8% of them using open systems exclusively. Retail customers were the least likely to use open systems only (17.3%), with smoke shop and internet customers in between (47.6% and 51.3%, respectively). *P*-values for differences between retail and each of the other three channels were all <0.001. The majority of retail customers used closed systems exclusively—76.4%, compared to 41.1% of smoke shop, 36.1% of internet, and 3.2% of vape shop customers (*p*-values of differences relative to retail all <0.001).

The bottom half of Table 2 presents the results for those who reported being smokers 12 months before the survey. Similar differences emerged in terms of the percentage of daily e-cigarette use and choice of models across the four purchase channels. For example, vape shop customers were more likely to be daily e-cigarette users (53.6% compared to 14.2% of retail customers). Vape shop customers were also more likely to use open systems (92.7% compared to 13.7% of retail customers).

Among those who reported being a smoker 12 months before the survey, there was no significant difference in the likelihood of being daily smokers when they were smoking 12 months ago. Approximately three-quarters of each group were daily smokers 12 months before the survey. This is important for the analysis presented below, which focuses on the quitting activities of these e-cigarette users.

Figure 1 presents rates of quit attempt and of smoking cessation for those who were smokers 12 months before the survey. As the left panel shows, the majority of these e-cigarette users made a quit attempt in the last 12 months. The percentage of users making a quit attempt was lowest among smoke shop customers (50.6%) compared to other groups, although only the difference between smoke shops and vape shops (66.4%) is statistically significant (*p* < 0.05). Quit attempt percentages were not statistically different among the three other groups (retail, internet, and vape shop customers).

The right panel shows the percentage of smokers who reported they quit smoking by the time of the survey. Vape shop and internet customers quit smoking at higher rates (22.2% and 22.5%, respectively) relative to retail and smoke shop customers (10.7% and 9.4%, respectively). Using the quit rate for retail customers as the reference point, the difference was statistically significant when comparing internet against retail (*p* = 0.025) and when comparing vape shop against retail (*p* = 0.014). Adjusting for demographics of users does not change the results, with *p*-values for the two comparisons at 0.022 and 0.010, respectively. The difference between smoke shop and retail customer quit rates was insignificant (*p* = 0.97).

Table 3 presents data on usage of FDA-approved quitting aids and of e-cigarettes in the last quit attempt reported in Figure 1. Retail customers were most likely to have used an FDA-approved quit aid, 46.9%, compared to smoke shop (23.5%, *p* < 0.01), and vape shop (21.8%, *p* < 0.001) customers. Vape shop customers, on the other hand, were the most likely to have used an e-cigarette—93.8%, followed by internet customers (89.6%), smoke shop (80.2%), and then retail (72.3%) customers (*p*-values for the differences between retail and internet, and retail and vape shop, were <0.05; *p*-value for the difference between retail and smoke shop customers was 0.30).

NRT was the most popular quitting aid. Some users used more than one quitting aid, but the difference across these four groups was mainly due to the difference in likelihood of using NRT.

It is worth noting that the 2016 survey asked those who smoked 12 months ago whether they had ever used FDA-approved quitting aids to help them with quitting smoking, regardless of whether the quit attempt occurred in the last 12 months. A majority (59.0%) of these respondents reported ever use of a FDA-approved quitting aid. The ever use rates of quitting aids among those who smoked 12 months ago were 62.0%, 58.4%, 58.0%, and 56.8%, for retail store, smoke shop, internet, and vape shop customers, respectively. There was no statistically significant difference among these four groups of customers (*p* = 0.94).

Finally, we compared the primary purchase channel for respondents in 2014 versus 2016 in Figure 2. In 2014, the most likely channel for e-cigarette purchase was general retail store (35.0%), followed by smoke shops (23.1%), internet (21.5%), and vape shops (20.4%). In 2016, the percentages for the first three channels dropped—general retail store to 27.7%, smoke shops to 17.1%, and internet to 17.5%. Meanwhile, vape shops grew substantially as a purchase channel (to 37.6%) and replaced general retail stores as the most likely channel for e-cigarette purchases in 2016.

## 4. Discussion

The study found e-cigarette users differed in vaping and smoking cessation patterns, depending on where they primarily purchased e-cigarette products. There are multiple potential reasons for these differences which our study cannot distinguish between. For example, differences could be the result of a self-selection process such that serious users tend to purchase e-cigarettes from vape shops, while less serious users tend to purchase from retail stores. The differences could also be due to interaction between users and products such that users who purchased certain types of products may have primarily purchased from certain types of outlets, and the products in turn affected their e-cigarette usage patterns.

Vape shop and retail store customers exhibited the sharpest differences in e-cigarette use patterns. Preference for e-cigarette device was strikingly different: 92.8% of vape shop customers used open-system models, compared to only 17.3% of retail customers. This preference for open systems also correlated with the frequency of vaping: more than half of vape shop customers used e-cigarettes daily, while only 1 of 5 retail customers did. Daily use of e-cigarettes has been found to be associated with smoking cessation [17,18,19], as has use of open designs [20,21,22]. It is thus expected that these groups would likely differ in their smoking cessation rate. That is, because vape shop customers were more likely to use open system e-cigarettes and more likely to vape on a daily basis relative to retail store customers, these differences could have contributed to differences in smoking cessation outcomes across purchase channels.

Vape shop and retail store customers indeed differed in smoking cessation outcomes, with the former more likely to have quit smoking in a 12-month period. One potential explanation for this difference in smoking cessation is that vape shop customers were simply more motivated to quit smoking than retail store customers. However, data shown in Figure 1 and Table 3 suggest that may not be the whole story. If “making a quit attempt” is used as a measure of motivation, vape shop smokers were not significantly more motivated than retail store customers (Figure 1). If using FDA-approved cessation aids in their most recent quit attempt is considered an indication of motivation to stop smoking cigarettes, then retail customers were more motivated than vape shop customers: retail store customers were twice as likely to use FDA-approved quitting aids as vape shop customers (Table 3). In other words, neither measure suggests the difference in smoking cessation between these two groups was entirely the result of differences in users’ initial motivation to quit smoking.

The fact that the vape shop and retail store customers did not differ in their rate of “having ever used FDA-approved quitting aids” also suggests that the vape shop customers were not inherently biased against using FDA-approved quitting aids. However, these two groups differed in their preference for using e-cigarettes rather than traditional quitting aids in their most recent quit attempt. Prior research has found higher quit rates through the use of e-cigarettes, as compared to rates for smokers using NRT or smoking cessation medications [23,24]. This suggests differences in use of e-cigarettes as a quit aid could have contributed to differences in smoking cessation outcomes by purchase channel. Relatedly, differences in the level of guidance and social support provided through vape shops versus other channels may also be contributing factors to differences found [5,7,8,9,10,11,12,25]. Customers of vape shops report professional guidance and support as reasons for purchasing from vape shops rather than through other channels [26]. Further, a pilot study found evidence of high success rates in quitting when first-time e-cigarette users were provided professional and technical support through vape shops [27]. It is possible that vape shop customers may be more likely to seek personalized guidance in the use of e-cigarettes in quit attempts than customers of other purchase channels, contributing to higher quit rates.

Another possibility for the higher smoking cessation rate for vape shop customers is that these customers were less addicted to cigarette smoking than retail store customers before they attempted to quit. However, these two groups did not differ significantly in their rate of daily cigarette usage 12 months prior to the survey. In other words, any potential difference in nicotine addiction level was not sufficiently large to explain the significant difference in cessation outcomes.

To some extent, these explanations can also be applied to internet customers. Internet customers were similar to vape shop customers in many ways—a finding in line with other research studying usage patterns of online e-cigarette customers [28]. In contrast, smoke shop customers were more like those who purchased e-cigarettes primarily from retail stores (Figure 1).

It is important to note that vape shops could present public health dangers. Researchers have raised the possibilities that vape shops encourage youths who have never smoked to experiment with e-cigarettes [29,30,31]. While there is general agreement about the lower toxicity of e-cigarettes relative to cigarettes, e-cigarettes still contain harmful substances [32]. There is also concern that e-cigarettes could be a “gateway” to cigarettes [33,34,35]. Interestingly, this study found vape shop e-cigarette users had the lowest percentage of never smokers as well as the lowest percentage of young adults age 18–24. We also found that vape shop users were more likely to (i) be former smokers and (ii) to report using an e-cigarette in their most recent quit attempt than retail and smoke shop customers, suggesting that vape shops may be particularly attractive to smokers who are interested in e-cigarettes as a method for smoking cessation. Overall, our study points to the need for further investigation of the role vape shops may play in tobacco control.

Finally, this study found there was a general market shift from retail stores to vape shops from 2014 to 2016. This is consistent with Braak et al., who found vape shops to be the most common purchase channel among U.S. vapers in 2016 [4]. We do not study the factors driving this change in demand towards vape shops. We speculate that one potential driver may be shifting consumer preferences from closed-system disposable ENDS to more customizable open-systems vaping devices [36]. From 2013–2014 to 2015–2017, e-cigarette brand websites increasingly offered open-systems designs, with increasing range in flavors and nicotine options [37]. Research also finds that, as e-cigarettes users become increasingly experienced, they are likely to transition from simple closed-system to newer generation advanced designs [21]. Thus, as the U.S. market has matured, both new and existing customers may be moving towards open-systems. This may have contributed to a shift in sales towards vape shops, which tend to focus on the sale of newer generation open-system product models [37].

There are two implications of this trend. First, it suggests limitations to reliance on Nielsen data about market trends in the e-cigarette industry, since Nielsen does not track sales through non-traditional channels such as the internet and specialty vape shops [2]. Focusing purely on Nielsen-covered channels may limit researchers’ ability to accurately describe vaper behavior. By 2016, less than 30% of vapers reported their primary place of purchase as retail stores, the only category covered by Nielsen. Gaining a more accurate description of market trends requires supplemental information about sales through channels not tracked by Nielsen as well as clear distinctions between different types of channels (such as smoke versus vape shops).

Second, given the association between higher quit rates and vape shops (relative to retail stores), this shift from retail store to vape shops could be viewed as a positive trend. Moreover, vape shops tend to be independently operated, while retail stores tend to be part of a larger network that is controlled by large tobacco companies [38]. However, the full implementation of the Deeming Rule may change the trajectory of current market trends. The Deeming Rule requires e-cigarette manufacturers to complete a labor- and resource-intensive application process that includes detailed ingredient, manufacturing, and product labeling/marketing information [39]. This could constrain vape shops’ ability to appeal to vaping enthusiasts by offering a variety of open-system designs and e-liquids. Vape shops are also themselves subject to new retailer requirements [40]. The findings from this study suggest that consideration of how regulations may ultimately shape e-cigarette users’ access to different purchase channels is warranted.

Finally, a number of vape shop retailers have demonstrated gaps in awareness of as well as weak compliance with impending FDA regulations. Yu et al. find a heavy reliance among vape shop employees on social media and e-cigarette company sales representatives for information about e-cigarette products [41]. The rising prominence of vape shops as a channel for e-cigarette purchases suggests they will play an increasingly important role in the extent to which FDA policies will be upheld within the e-cigarette marketplace. They also will increasingly influence the type and quality of information conveyed to e-cigarette users. Our study suggests the need for stronger FDA communication efforts about e-cigarette rules and policies as well as greater training and monitoring for compliance.

## 5. Limitations

There are several limitations for the estimates presented in this study. First, they were based on users’ self-report, which may be biased. Relatedly, respondents were asked where they usually purchased their products, although respondents may still have purchased e-cigarette products through other channels. What we present should thus be viewed as rough estimates of the relative proportion of e-cigarette users’ primary place of purchase. These data may be viewed as a first step towards complementing the data available about e-cigarette sales through general retail stores such as Nielsen datasets. A third limitation is that the results about smoking cessation and place of purchase were correlational, as purchase behavior is based on self-selection, which is not readily subjected to controlled trials.

## 6. Conclusions

Our study points to a need for policymakers to consider how regulatory implementation may relate to the different channels through which consumers purchase electronic nicotine delivery systems. Recently, the FDA has proposed a policy framework for sale of flavored ENDS products that restricts sales of most flavors to channels that only sell to adults, such as vape shops and online retailers, or to general retail stores that segregate ENDS flavors to an adults-only section [42]. This regulatory move could shift the distribution of sales across channels, perhaps increasing vape shop shares further. Better understanding of how such regulatory changes may affect overall e-cigarette user behaviors requires better understanding of how user behaviors relate to e-cigarette channel of purchase.

## Figures and Tables

**Figure 1 ijerph-16-00724-f001:**
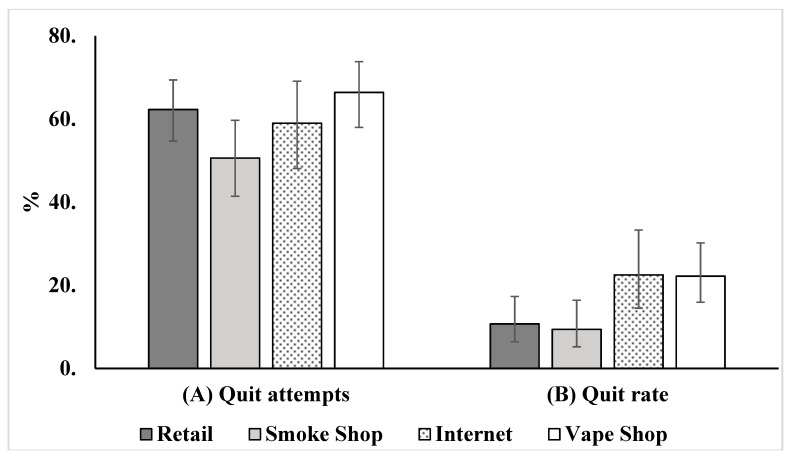
Quit attempt and cessation rate by place of purchase, among those who were smokers 12 months before the survey. (**A**) Quit attempts; (**B**) quit rate.

**Figure 2 ijerph-16-00724-f002:**
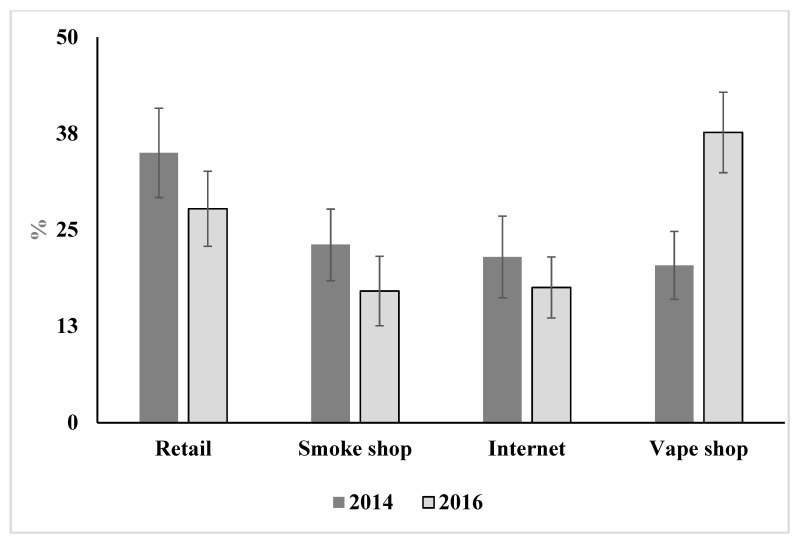
The changing market share, based on e-cigarette users’ self-report, from 2014 to 2016.

**Table 1 ijerph-16-00724-t001:** Demographics of e-cigarette users by place of purchase (N = 1622, with 2014 and 2016 surveys combined).

	Retail % (95% CI)	Smoke Shop % (95% CI)	Internet % (95% CI)	Vape Shop % (95% CI)
**Gender**	N = 532	N = 339	N = 308	N = 443
Male	43.9 (37.0–51.0)	52.2 (43.8–60.6)	43.5 (35.4–51.9)	47.8 (41.1–54.6)
Female	56.1 (49.0–63.0)	47.8 (39.4–56.2)	56.5 (48.1–64.6)	52.2 (45.4–58.9)
**Age**				
18–24	12.0 (7.6–18.6)	12.9 (7.4–21.5)	15.5 (10.5–22.1)	3.2 (1.5–6.7)
25–34	24.5 (18.8–31.3)	24.0 (17.6–31.8)	29.4 (22.0–38.1)	39.3 (32.8–46.1)
35–44	22.6 (17.3–28.9)	13.6 (8.6–20.8)	18.5 (12.9–25.9)	21.8 (16.5–28.1)
45–54	20.4 (15.5–26.5)	21.7 (15.7–29.2)	15.5 (10.1–23.0)	15.9 (11.4–21.7)
55–64	12.6 (8.8–17.7)	19.5 (13.9–26.7)	14.4 (9.5–21.1)	14.4 (10.3–19.9)
65+	7.5 (4.7–11.9)	7.9 (4.6–13.2)	6.6 (3.5–11.8)	5.3 (3.0–9.1)
**Education**				
Low	51.7 (44.6–58.8)	60.8 (52.3–68.6)	55.4 (47.0–63.5)	49.9 (43.1–56.7)
High	48.3 (41.2–55.4)	39.2 (31.4–47.7)	44.6 (36.5–53.0)	50.1 (43.3–56.9)
**Race**				
NH–White	71.8 (65.0–77.8)	66.7 (58.2–74.2)	67.4 (59.0–74.8)	75.1 (68.5–80.6)
NH–Black	8.3 (5.2–12.9)	10.0 (6.2–15.8)	1.7 (0.3–10.4)	8.9 (5.8–13.5)
Hispanic	11.6 (7.6–17.4)	18.0 (12.4–25.4)	20.7 (14.9–28.0)	8.0 (4.9–12.8)
NH–Other	8.1 (4.9–13.1)	5.1 (2.1–11.7)	9.7 (5.4–16.7)	7.9 (4.7–13.0)

**Table 2 ijerph-16-00724-t002:** Cigarette smoking behavior and the use of e-cigarette systems (2014 and 2016 surveys combined).

	Retail	Smoke Shop	Internet	Vape Shop
**All E-cigarette Users at the Time of Survey**	N = 532 % (95% CI)	N = 339 % (95% CI)	N = 308 % (95% CI)	N = 443 % (95% CI)
Smoking Status				
Never smokers	8.2 (3.8–16.8)	10.1 (5.5–17.8)	16.1 (11.0–23.0)	5.3 (3.1–9.1)
Current smokers	73.7 (66.5–79.8)	70.4 (61.4–78.0)	48.7 (40.3–57.1)	54.5 (47.4–61.3)
Former smokers	17.7 (12.5–24.4)	19.3 (13.0–27.7)	35.1 (27.5–43.7)	40.2 (33.5–47.3)
Frequency of Current E-cigarette Use				
Daily use	19.7 (14.4–26.4)	23.2 (16.8–31.3)	42.9 (34.8–51.4)	59.1 (52.2–65.7)
Non–daily Use	80.3 (73.6–85.6)	76.8 (68.7–83.2)	57.1 (48.6–65.2)	40.9 (34.3–47.8)
E-cigarette Models				
Open system	17.3 (12.7–23.1)	47.6 (39.2–56.1)	51.3 (42.7–59.8)	92.8 (87.8–95.8)
Closed system	76.4 (69.8–81.9)	41.1 (33.1–49.5)	36.1 (28.4–44.7)	3.2 (1.5–6.7)
Both	6.2 (3.3–11.3)	11.3 (7.0–17.8)	12.5 (8.1 – 18.9)	3.9 (1.7–8.6)
**E-cigarette Users Who were Smoking Cigarettes 12 Months Ago**	N = 459 % (95% CI)	N = 302 % (95% CI)	N = 238 % (95% CI)	N = 314 % (95% CI)
Smoking Behavior 12 Months Ago				
Daily smokers	75.2 (67.4–81.6)	75.0 (65.9–82.3)	72.8 (62.0–81.5)	78.4 (70.0–85.0)
Occasional smokers	24.8 (18.4–32.6)	25.0 (17.7–34.1)	27.2 (18.5–38.0)	21.6 (15.0–30.0)
Frequency of Current E-cigarette Use				
Daily Use	14.2 (9.3 – 21.2)	19.8 (13.4–28.2)	33.0 (23.6–43.9)	53.6 (45.0–62.0)
Non–daily Use	85.8 (78.8 – 90.7)	80.2 (71.8–86.6)	67.0 (56.1–76.4)	46.4 (38.0–55.0)
E-cigarette Models				
Open system	13.7 (9.4–19.6)	46.9 (37.9–56.1)	54.7 (43.7–65.2)	92.7 (86.8–96.1)
Closed system	79.0 (71.9–84.7)	45.0 (36.1–54.2)	38.8 (29.0–49.7)	3.7 (1.6–8.3)
Both	7.1 (3.7–13.4)	8.0 (4.1–15.2)	6.4 (2.7–14.2)	3.5 (1.4–8.9)

**Table 3 ijerph-16-00724-t003:** Use of FDA-approved quitting aids or E-cigarettes in the last quit attempt, among those who smoked 12 months ago.

For Those Who Made a Quit Attempt	Retail N = 259 % (95%CI)	Smoke Shop N = 136 % (95%CI)	Internet N = 129 % (95%CI)	Vape Shop N = 189 % (95%CI)
Any Quitting Aid	46.9 (37.0–57.1)	23.5 (14.0–36.5)	30.2 (19.2–44.2)	21.8 (14.3–31.7)
NRT	42.5 (32.7–52.9)	17.3 (9.2–30.2)	22.9 (13.4–36.2)	18.0 (11.2–27.7)
Zyban	12.6 (6.9–21.8)	5.5 (1.9–15.2)	12.0 (5.8–23.3)	3.9 (1.5–10.0)
Chantix	10.2 (5.2–19.1)	6.0 (1.9–17.3)	14.8 (7.4–27.3)	6.5 (3.0–13.4)
E-cigarettes	72.3 (61.9–80.7)	80.2 (67.1–88.9)	89.6 (78.0–95.4)	93.8 (83.4–97.8)

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
