# Peer review of "A Comparison of E-Cigarette Use Patterns and Smoking Cessation Behavior among Vapers by Primary Place of Purchase"

_ijerph, 2019, doi:10.3390/ijerph16050724_

Round 1
Reviewer 1 Report
This is a very nice paper that addresses two important issues. First, where do vapers purchase their electronic cigarettes? Second, does purchase location for vaping products influence the types of products purchased and smoking cessation. The study makes use of a large representative sample of current e-cigarette users. The findings from the paper are consistent with a recently published paper in IJERPH which examined the question of where current vapers reported purchasing their electronic cigarette products and devices in different countries (US, Canada, Australia). I would encourage the authors to reference this paper and comment on the similarity/differences of results between their study and this recently report study. The citation to this study is shown below:
Braak D, Cummings KM, Nahhas GJ, Heckman BW, Borland R, Fong GT, Hammond D, Boudreau C, McNeill A, Levy DT, Shang C. Where do vapers buy their vaping supplies? Findings from the International Tobacco Control (ITC) 4 Country Smoking and Vaping Survey. 2019. Int. J. Environ. Res. Public Health 2019, 16, 338; doi:10.3390/ijerph16030338
One of the important findings of this study is the observation that those purchasing their electronic cigarettes from vapes shops and/or the internet were more likely to have stop smoking completely. This finding is consistent with two other published studies that the authors should reference. The citations to these two studies are shown below:
Polosa, R.; Caponnetto, P.; Cibella, F.; Le-Houezec, J. Quit and Smoking Reduction Rates in Vape Shop Consumers: A Prospective 12-Month Survey. Int. J. Environ. Res. Public Health 2015, 12, 3428–3438.
Wagener, T.L.; Shaikh, R.A.; Meier, E.; Tackett, A.; Tahirkheli, N.; Leavens, E.; Driskill, L. Examining the Smoking and Vaping Behaviors and Preferences of Vape Shop Customers. Tob. Prev. Cessation. 2016,
This study shows a major shift in the percentage of current e-cigarette users purchasing from retail establishments in 2014 to vape shops in 2016. It would be helpful if the authors could comment on why they think this shift has taken place. How might changes in the price of electronic cigarettes sold in retail stores compared to vape shops have influence this shift in purchase patterns.
The authors have longitudinal data for current vapers surveyed in 2014 and surveyed again in 2016. Rather than examine these data the authors have chosen to randomly assign respondents who participated in both surveys to either the 2014 or 2016 surveys so that cross-sectional estimates on vaping location in each year can be compared. While this approach is ok for the cross-sectional comparison, it would be valuable to also examine how vaping behaviors changed in those surveyed in both 2014 and 2016. How many current vapers in 2014 were still vaping in 2016? What factors predict continued vaping? Do respondents who continue to vape report purchasing in the same or different locations? Do vapers report using the same or different types of products? While I recognize that some of these findings likely could form the basis for another paper, it would be helpful for the readers of this paper to get some sense of longitudinal stability of vaping behavior and purchasing patterns. I suspect that most of those who reported being current vapers in the 2014 were likely not vaping in 2016, which would preclude looking at how purchase patterns changed overtime. However, if that is the case, this would be an important finding and ought to be reported. The authors might consider referencing the paper by Kasza et al below that looked at longitudinal transitions in use of tobacco products, including electronic cigarettes.
Kazsa KA et Transitions in Tobacco Product Use by U.S. Adults between 2013⁻2014 and 2014⁻2015: Findings from the PATH Study Wave 1 and Wave 2. Int J Environ Res Public Health. 2018 Nov 9;15(11). pii: E2515. doi: 10.3390/ijerph15112515
Reviewer 2 Report
Thank you for this great manuscript, indeed looking into a less familiar aspect of the booming e-cigarette business, e-cig purchase place relating to e-cig use and cig cessation.
Minor comments:
Line 81: Would it be possible to add something along the lines of: the GfK (established in 1934 as Gesellschaft für Konsumforschung, German for "Society for Consumer Research") -- for those unfamiliar with GfK.
Line 86: Is there a reference for the statement in parentheses?
Line 135: Maybe define NRT as nicotine replacement therapy, again as a formality.
Line 169: There seems to be an extra 7 at the end of the sentence.
Lines 204-210: For this paragraph, is there also a clarification that the quit rate for retail customers was used as a reference point, similar to the paragraph following after?
Line 256: Perhaps "reasons for these differences between which our study cannot distinguish."
Lines 286-288: Could you possible expand on how the preference difference (e-cigs vs traditional quitting aids) may have affected quit attempts? What about e-cigs and vapers using those devices made them more likely to quit? Is that a view shared by most vapers or does it vary from user to user? Is that significant?
Lines 308-314: Was Nielsen data previously mentioned? Just seems slightly abrupt to mention for the first time.
Reviewer 3 Report
The conclusion section is very small. Authors should to expand this sectio describing the scientifical and practica impact of the results.
The literature section should be developed and contain more peer reviewed international journals to compare the results.
